

# Association between gonadal hormones and osteoporosis in schizophrenia patients undergoing risperidone monotherapy: a cross-sectional study

Yi Chen[1], Yaoyao Zhang[2], Kaili Fan[2], Weiqian Xu[3], Chao Teng[4], Shuangshuang Wang[5], Wei Tang[1,2] and Xiaomin Zhu[5]

[1] Department of Psychiatry, The Affiliated Kangning Hospital of Wenzhou Medical University, Wenzhou, Zhejiang, China
[2] Department of Psychiatry, Wenzhou Medical University, Wenzhou, Zhejiang, China
[3] Department of Psychiatry, The Second People's Hospital of TaiZhou, Taizhou, Zhejiang, China
[4] Department of Psychiatry, Zhejiang Chinese Medical University, Hangzhou, Zhejiang, China
[5] Department of Psychiatry, Suzhou Guangji Hospital, the Affiliated Guangji Hospital of Soochow University, Suzhou, Jiangsu, China

Corresponding author
Xiaomin Zhu, lizamin@163.com

## ABSTRACT

**Objective**. Patients with schizophrenia are at increased risk of osteoporosis. This study first determined the osteoporosis rate in patients with schizophrenia and then then explored the association between serum gonadal hormone levels and osteoporosis among these patients.

**Methods**. A total of 250 patients with schizophrenia and 288 healthy controls were recruited. Osteoporosis was defined by decreased bone mineral density (BMD) of the calcaneus. Serum fasting levels of gonadal hormones (prolactin, estradiol, testosterone, progesterone, follicle-stimulating hormone, luteinizing hormone) were determined. The relationship between osteoporosis and hormone levels was statistically analyzed by binary logistic regression analysis.

**Results**. Our results showed that patients with schizophrenia had a markedly higher rate of osteoporosis (24.4% vs. 10.1%) than healthy controls ($P < 0.001$). Patients with osteoporosis were older, had a longer disease course, and had a lower body mass index (BMI) than patients without osteoporosis (all $P < 0.05$). Regarding gonadal hormones, we found significantly higher prolactin, but lower estradiol, levels in patients with osteoporosis than in those without osteoporosis (both $P < 0.05$). The regression analysis revealed that PRL (OR = 1.1, 95% CI [1.08–1.15], $P < 0.001$) and E2 level (OR = 0.9, 95%CI [0.96–0.99], $P = 0.011$) were significantly associated with osteoporosis in patients with schizophrenia.

**Conclusion**. Our results indicate that patients with schizophrenia who are being treated with risperidone have a high rate of osteoporosis. Increased prolactin and reduced estradiol levels are significantly associated with osteoporosis.

## INTRODUCTION

Osteoporosis is a degenerative disease that is characterized by a decrease in bone mineral density (BMD) and results in an increased risk of fractures (*Liang et al., 2019*). Approximately 200 million people worldwide are affected by osteoporosis, which increases their morbidity and mortality (*Cui et al., 2018*). Numerous studies have been conducted to explore risk factors for osteoporosis, and commonly reported risk factors are old age, female sex, insufficient calcium intake, inadequate physical activity, excessive smoking, excessive drinking and use of antipsychotics (*Li et al., 2017*; *Crews & Howes, 2012*).

Schizophrenia is a severe, chronic, and debilitating disorder that affects approximately 1% of the global population (*Stępnicki, Kondej & Kaczor, 2018*). Antipsychotic drugs are considered the primary treatment for schizophrenia. Although these drugs have significant benefits for psychotic symptoms, they can induce health problems such as metabolic syndrome, cardiovascular diseases, sexual dysfunction, and osteoporosis (*Andrade, 2016*). Previous studies have demonstrated that patients with schizophrenia have a higher risk of osteoporosis than the general population (*Cui et al., 2018*; *Stubbs et al., 2015*).

The underlying mechanisms of increased osteoporosis risk in patients with schizophrenia are still unclear. However, studies have shown that, in addition to poor nutrition, reduced physical activity, excessive smoking, and drinking, the main reason that patients with schizophrenia develop osteoporosis is the use of antipsychotic drugs (*Li et al., 2017*; *Halbreich et al., 2003*). Antipsychotics can elevate the secretion of prolactin (PRL) via the dopamine D2 receptor-blocking effect (*Peuskens et al., 2014*). In addition, hyperprolactinemia caused by antipsychotics can lead to estrogen and androgen deficiencies, which can accelerate bone loss and increase the risk of osteoporosis (*Okita et al., 2014*). A large body of evidence supports that decreased estrogen levels can increase bone resorption by prolonging the life of osteoclasts (*Li et al., 2018*; *Chopko & Lindsley, 2018*) and that androgen deficiency can lead to an imbalance of osteoblast and osteoclast activity, resulting in decreased osteogenesis (*Mohamad, Soelaiman & Chin, 2016*). Taken together, these findings suggest that abnormal levels of gonadal hormones due to the use of antipsychotic drugs could be associated with osteoporosis in patients with schizophrenia. To further explore this hypothesis, we focused on schizophrenia patients using a single antipsychotic drug, risperidone, which is commonly used in our clinical practice.

In clinical practice, risperidone is a widely used second-generation antipsychotic (SGA), as well as a prolactin-elevating compound (*Bishop et al., 2012*). Early studies of patients treated with risperidone found high PRL levels to be associated with low BMD values (*Becker et al., 2003*). However, other studies failed to replicate this pattern (*Stubbs et al., 2014*). The inconsistent results may be related to confounding factors, such as age, gender and varying levels of physical activity.

In the present study, we aimed to analyze risk factors associated with osteoporosis in inpatients with schizophrenia receiving risperidone monotherapy. We speculate that abnormal gonadal hormone levels may be associated with the onset of osteoporosis in patients with schizophrenia. We hope that this work can provide clinical data on the

osteoporosis rate and its associated risks in patients with schizophrenia, which helps us to take measures to prevent osteoporosis and treat it.

## MATERIALS AND METHODS

### Participants

A total of 250 inpatients with schizophrenia who were hospitalized in Kangning Hospital Affiliated to Wenzhou Medical University from May 2018 to June 2020 were included in our study. All patients met the following criteria: (1) a diagnosis of schizophrenia according to the Diagnostic and Statistical Manual of Mental Disorders, Fourth Edition (DSM-IV) (*American Psychiatric Association, 1994*); (2) age 18–75 years old; (3) Han Chinese ethnicity; and (4) undergoing risperidone monotherapy for at least 6 months. The exclusion criteria were as follows: (1) diagnosis of a psychiatric disorder in addition to schizophrenia or a history of substance abuse/dependence disorder; (2) severe cardiovascular, hepatic, or renal diseases that may affect bone metabolism, such as diabetes or hyperthyroidism; (3) pregnancy or breastfeeding; and (4) a history of bone fracture within one year before enrollment. Since admission to the facility, all patients had followed the same diet and activity schedule, which helps to minimize differences in physical exercise and diet between patients. According to the principles of frequency matching, we selected a group of 288 healthy controls from the physical examination center at our hospital. The control group was comparable to the patient group in terms of age and sex. This study was performed in strict accordance with the Declaration of Helsinki and all other relevant national and international regulations. The study protocol was approved by the Medical Ethics Committee of The Affiliated Kangning Hospital of Wenzhou Medical University (approval number: 20180412001). All participants signed informed consent before the formal study. Written informed consent was obtained from all participants prior to their participation in any procedures related to this study.

### Assessment of participant characteristics

Detailed demographic and clinical data were collected via a standardized form that was specifically designed for this study. Weight and height were measured in a standardized manner. Participants were barefoot and stood upright, while height was measured to the nearest millimeter. An electronic scale was used to evaluate weight while wearing light indoor clothing. Body mass index (BMI) was calculated as weight in kg/square of height in meters.

### Definition of osteoporosis

BMD ($g/cm^2$) of the calcaneus was measured by a trained ultrasound technician blind to our research hypotheses in a separate examination center at the hospital using a 3.01 Sahara Clinical Bone Sonometer (Hologic). Quantitative ultrasound of the calcaneus (QUS) measurements were performed at the right heel (or left heel, if the right heel was inaccessible) using a 6 Broadband ultrasound attenuation (BUA; DB/MHz) and speed of sound (SOS; m/s) at least twice on each calcaneum. The T-score of QUS refers to the number of standard deviations (SD) away from the mean T-score of a database of normal

values compiled from a healthy young adult population and was calculated as $(0.67 \times BUA + 0.28 \times SOS) - 420$. According to World Health Organization (WHO) criteria (World Health Organization Study Group, 1994), 0-steoporosis is defined by BMD.

### Measurement of serum gonadal hormones

A 10-ml fasting blood sample was drawn from each patient between 6:00 and 9:00 a.m. and stored in ice-cooled ethylenediaminetetraacetic acid tubes. Serum was separated by centrifugation at 5°C and stored at $-20$ °C. Gonadal hormones, including PRL, estradiol (E2), testosterone (T), progesterone (P), follicle-stimulating hormone (FSH), and luteinizing hormone (LH), were measured using the chemiluminescence immunoassay kits ARCHITECT and ARCHITECT i2000 (Abbott Japan Co., Chiba, Japan).

### Statistical analysis

Statistical analyses were performed using SPSS software version 26.0 (SPSS, Chicago, IL). First, we used independent samples t-tests or the Mann-Whitney U test for continuous variables and the chi-square test for categorical variables to compare differences between groups. Second, using osteoporosis as the dependent variable and age, BMI, total disease course, estradiol level, prolactin level, FSH level and risperidone doses as independent variables, we performed a binary logistic regression analysis with the "enter" method to identify factors independently associated with osteoporosis in patients with schizophrenia. All tests were two-tailed, and the statistical significance level was set as $P \leq 0.05$.

## RESULTS

### Demographic characteristics of patients and controls

The demographic data of the participants are presented in Table 1. For patients with schizophrenia, the average age was $46.3 \pm 10.6$ years, and the average BMI was $24.5 \pm 4.0$ kg/m$^2$. For healthy controls, the average age was $46.0 \pm 10.8$ years, and the average BMI was $23.4 \pm 3.5$ kg/m$^2$. There were no significant differences in sex or age between patients and controls (age: $P = 0.613$; sex: $P = 0.101$). Patients had a significantly higher BMI than healthy controls ($P = 0.007$). Among patients with schizophrenia, the average disease duration was $21.0 \pm 9.0$ years.

### Rates of osteoporosis in patients and controls

The rates of osteoporosis were 24.4% (61/250) for patients with schizophrenia and 10.1% (29/288) for healthy controls. The patient group had a significantly higher rate of osteoporosis than the control group ($P < 0.001$).

### Comparison of demographic and clinical characteristics and gonadal hormone levels between schizophrenia patients with and without osteoporosis

Our results showed that schizophrenia patients with osteoporosis were older than those without osteoporosis ($51.7 \pm 11.6$ vs $44.6 \pm 9.7$, $Z = -4.6$, $p < 0.001$). Moreover, patients with osteoporosis had a longer duration of illness ($26.2 \pm 9.3$ vs $19.5 \pm 8.8$, $Z = -4.9$, $P < 0.001$) and lower BMI ($23.1 \pm 3.06$ vs $24.9 \pm 4.1$, $t = 3.5$, $P < 0.001$). There were no

**Table 1  Comparison between schizophrenia patients and control subjects.**

|  | Patients ($N = 250$) | Controls ($N = 288$) | t/Z/X$^2$ | P |
|---|---|---|---|---|
| Age (years) | $46.3 \pm 10.6$ | $46.0 \pm 10.8$ | $-0.51$ | 0.613 |
| Sex (male/female) |  |  |  | 0.101 |
| Male | 137 | 137 |  |  |
| Female | 113 | 151 |  |  |
| Body mass index (Kg/m$^2$) | $24.5 \pm 4.0$ | $23.4 \pm 3.5$ | $-2.704$ | 0.007 |
| Total disease courses (years) | $21.0 \pm 9.0$ | – |  |  |
| Bone mass density (g/cm$^2$) | $0.5 \pm 0.1$ | $0.5 \pm 0.1$ | $-3.60$ | <0.001 |
| Osteoporosis |  |  |  | <0.001 |
| Yes | 61 | 29 |  |  |
| No | 189 | 259 |  |  |

Notes.
BMI, body mass index; BMD, bone mass density.

**Table 2  Differences between schizophrenia patients with osteoporosis and without osteoporosis.**

|  | Patients with osteoporosis $N = 61$ | Patients without osteoporosis $N = 189$ | t/Z/X$^2$ | P |
|---|---|---|---|---|
| Age (year) | $51.7 \pm 11.6$ | $44.6 \pm 9.7$ | $-4.6$ | <0.001 |
| Sex (male/female) | 40/21 | 97/92 |  | 0.052 |
| Course of disease (years) | $26.2 \pm 9.3$ | $19.5 \pm 8.8$ | $-4.9$ | <0.001 |
| Body mass index (kg/m$^2$) | $23.1 \pm 3.1$ | $24.9 \pm 4.1$ | 3.5 | 0.001 |
| Daily dosage of risperidone | $4.8 \pm 1.7$ | $4.7 \pm 1.9$ | $-0.7$ | 0.466 |
| Estradiol (pmol/L) | $55.3 \pm 24.6$ | $88.0 \pm 45.3$ | $-5.3$ | <0.001 |
| Testosterone (nmol/L) | $9.2 \pm 5.5$ | $8.9 \pm 6.3$ | $-0.3$ | 0.730 |
| Progesterone (nmol/L) | $0.6 \pm 0.3$ | $0.7 \pm 0.5$ | $-1.0$ | 0.340 |
| Prolactin (ug/L) | $90.9 \pm 20.5$ | $44.1 \pm 17.2$ | $-10.6$ | <0.001 |
| Follicle-stimulating hormone (IU/L) | $7.6 \pm 4.1$ | $6.6 \pm 4.0$ | $-2.2$ | 0.026 |
| Luteinizing hormone (IU/L) | $9.0 \pm 3.5$ | $8.8 \pm 4.4$ | $-1.0$ | 0.302 |

Notes.
BMI, body mass index; BMD, bone mass density; E2, Estradiol; T, Testosterone; P, Progesterone; PRL, prolactin; FSH, Follicle-stimulating hormone; LH, luteinizing hormone.

significant differences in sex or daily risperidone dose between patients with and without osteoporosis (both $P > 0.05$). For gonadal hormone levels, patients with osteoporosis had a significantly higher PRL level ($90.9 \pm 20.5$ vs $44.1 \pm 17.2$, $Z = -10.6$, $P < 0.001$), a lower E2 level ($55.3 \pm 24.6$ vs $88.0 \pm 45.3$, $Z = -5.3$, $P < 0.001$) and a higher FSH level ($7.6 \pm 4.1$ vs $6.6 \pm 4.0$, $Z = -2.2$, $P = 0.026$) than patients without osteoporosis (Table 2).

## Factors associated with osteoporosis in patients with schizophrenia

The binary logistic regression analysis found that PRL levels were positively associated with osteoporosis in patients (OR = 1.1, 95% CI [1.08–1.15], $P < 0.001$), and E2 levels were negatively associated with osteoporosis in patients (OR = 0.9, 95% CI [0.96–0.99], $P = 0.011$) (see Table 3), accounting for 52% of the variance of osteoporosis in patients.

**Table 3  Results of the stepwise logistic regression analysis: independent risk factors for osteoporosis in schizophrenia patients.**

| Predictors | Sig | Odds ratio | 95%CI for Exp. (B) |
|---|---|---|---|
| Age(years) | 0.773 | 0.9 | 0.85–1.13 |
| Total disease courses(years) | 0.176 | 1.1 | 0.95–1.34 |
| Bone mass density (Kg/m$^2$) | 0.826 | 1.0 | 0.86–1.13 |
| Estradiol (pmol/L) | 0.011 | 0.9 | 0.96–1.00 |
| Prolactin (ug/L) | 0.000 | 1.1 | 1.09–1.15 |
| Follicle-stimulating hormone, (IU/L) | 0.407 | 1.1 | 0.92–1.22 |
| Daily dosage of risperidone | 0.407 | 1.1 | 0.66–1.19 |

Notes.
Total model: $P < 0.001$, R square: 0.520.
E2, Estradiol; PRL, prolactin.

# DISCUSSION

In the present study, we found that schizophrenia patients undergoing risperidone monotherapy had a higher rate of osteoporosis than healthy controls. We demonstrated that 24.4% of patients treated with risperidone had osteoporosis, which represents a 2.4-fold increased risk compared to healthy controls. Of note, a recent meta-analysis reported that osteoporosis is over 2.5 times more common in patients with schizophrenia treated with antipsychotics than in age- and sex-matched controls (*Stubbs et al., 2014*). In previous studies, the recruited patients used different antipsychotics, which may affect the rate of osteoporosis. When we only used risperidone monotherapy, the results were still consistent with the majority of previous studies (*Cui et al., 2018*; *Stubbs et al., 2014*; *Gomez et al., 2016*). To the best of our knowledge, this is the first study exploring the rate of osteoporosis in schizophrenia inpatients with risperidone monotherapy.

Despite some investigations, the precise role of antipsychotics in osteoporosis risk remains unclear. One potential mechanism relates to choice in the hypothalamic-pituitary-gonadal axis induced by antipsychotics (*Kishimoto et al., 2008*). The dopamine D2 receptor-blocking effect of antipsychotics could elevate the secretion of PRL, causing hyperprolactinemia (*Meaney et al., 2004*). The increased PRL level then leads to attenuation of the bone resorption rate, which consequently lowers the secretion of sex hormones such as E2 and T (*Cui et al., 2018*; *Okita et al., 2014*). In light of awareness that different types of antipsychotics have different effects on PRL, a recent meta-analysis and systematic review demonstrated that patients treated with PRL-raising antipsychotics (typical antipsychotics, risperidone, paliperidone, amisulpride) have a higher risk of BMD loss and osteoporosis than patients receiving PRL-decreasing antipsychotics (*Tseng et al., 2015*; *Lally et al., 2019*). Risperidone, which has difficulty penetrating the blood-brain barrier, is expected to have longer-lasting D2 antagonistic effects in the pituitary system than in the central nervous system, ultimately leading to prolonged hyperprolactinemia and maximal loss of BMD (*Markianos, Hatzimanolis & Lykouras, 2001*). Thus, risperidone may be the most likely to cause hormonal dysfunction, thereby increasing osteoporosis. Hence, clinical practice should be alert to the use of risperidone.

To date, no study has been conducted to explore the relationship between hypothalamic-pituitary-gonadal axis-related hormones and osteoporosis in schizophrenia patients receiving risperidone monotherapy. Our study provides new insights into the relationship between the two and further supports the role of gonadal hormones in the risk of osteoporosis in patients receiving risperidone. Specifically, we found that patients with osteoporosis had significantly higher PRL levels but lower E2 levels than patients without osteoporosis, which is similar to the findings of previous studies (*Liang et al., 2016*). Moreover, logistic regression analysis confirmed that PRL and E2 were independent predictive factors associated with osteoporosis after controlling for other known associated factors. Despite some research demonstrating significant correlations between P, FSH and TH (*Doumouchtsis et al., 2008*; *Prior, 2018*), we did not observe such relationships, which is in line with the majority of previous studies (*Aydin et al., 2005*; *Seven et al., 2016*). The association of low E2 levels with osteoporosis is consistent with the high incidence of osteoporosis in postmenopausal women (*Liang et al., 2016*; *Seven et al., 2016*) and supports the view that estrogen is involved in bone metabolism.

In addition, our study showed that schizophrenia patients with osteoporosis were older and had a longer total disease course than those without osteoporosis. These two risk factors have also been reported previously (*Cui et al., 2018*; *Kinon et al., 2013*; *Jung et al., 2006*). It is well known that the aging process increases bone destruction and decreases bone formation (*Tseng et al., 2015*). Patients with schizophrenia with a longer disease course may have received longer treatment with antipsychotics, thus resulting in more profound effects on gonadal hormones. However, our logistic regression analysis showed that age and disease course did not have significant clinical effects on osteoporosis. Rather, the results suggested that gonadal hormones, namely, PRL and E2, are important to osteoporosis risk. Nevertheless, the relationship between osteoporosis and disease course warrants further verification in prospective and longitudinal studies.

Unexpectedly, we found a lower BMI in patients with osteoporosis than in patients without osteoporosis. Previous research has reported a significant positive correlation between BMI and osteoporosis (*Liang et al., 2019*; *Bulut et al., 2016*). One possible reason for this relationship is that the higher BMI caused by SGA use could counteract BMD loss effects in schizophrenia (*Doknic et al., 2011*). Although risperidone showed a slight effect on body weight, in the logistic regression analysis, the difference between groups disappeared. Extensive longitudinal research with larger samples is needed to confirm these relationships.

The strength of this study is the relatively large sample of schizophrenia patients receiving risperidone monotherapy. However, there are several limitations worth mentioning. First, the cross-sectional nature of this research provides a limited capacity to identify a causal relationship between gonadal hormones and BMD loss or osteoporosis. Second, since all patients were inpatients recruited from one hospital in Wenzhou, our findings may not be generalizable to other settings and outpatient populations. Third, although all patients received risperidone monotherapy for at least six months, we did not collect detailed information about medication history, which may have effects on gonadal hormones. Fourth, we only recruited patients following a similar diet and physical exercise schedule,

which will cause selection bias to a certain extent. Finally, future research with a prospective and longitudinal design is required to evaluate causal relations between gonadal hormones and osteoporosis in first-episode and drug-naïve patients with schizophrenia.

In summary, our results provide further evidence of the increased rate of osteoporosis in patients with risperidone monotherapy compared to controls. Patients with osteoporosis tended to be older, have a longer disease course, have a higher BMI, have significantly higher PRL levels, and have lower E2 levels than patients without osteoporosis. Our binary regression logistic analysis showed that PRL and E2 levels were independently associated with osteoporosis. These findings suggest that PRL and E2 levels are related to osteoporosis in patients treated with risperidone. Prospective and longitudinal research is warranted to confirm these findings and investigate the underlying mechanism.

### Abbreviations

| | |
|---|---|
| **BMD** | Bone Mineral Density |
| **BMI** | Body Mass Index |
| **PRL** | Prolactin |
| **E2** | estradiol |
| **T** | Testosterone |
| **P** | Progesterone |
| **FSH** | follicle-stimulating hormone |
| **LH** | luteinizing hormone |
| **SGA** | Second Generation Antipsychotic |

## ACKNOWLEDGEMENTS

We are deeply grateful to all the patients with schizophrenia and healthy controls participating in this study as well as to the psychiatrists for their help in the recruitment and diagnosis of schizophrenic patients.

### Funding

The present work was supported by the Wenzhou Basic Medical Science and Technology Project (Y20190478), the Natural Science Foundation of Jiangsu Province (BK20180213), and the Basic Medical Science and Technology Project of Zhejiang (2020KY926). The funders had no role in study design, data collection and analysis, decision to publish, or preparation of the manuscript.

### Grant Disclosures

The following grant information was disclosed by the authors:
Wenzhou Basic Medical Science and Technology Project: Y20190478.
Natural Science Foundation of Jiangsu Province:  BK20180213.
Basic Medical Science and Technology Project of Zhejiang:  2020KY926.

## Competing Interests

The authors declare there are no competing interests.

## Author Contributions

- Yi Chen, Wei Tang and Xiaomin Zhu conceived and designed the experiments, authored or reviewed drafts of the paper, and approved the final draft.
- Yaoyao Zhang and Kaili Fan performed the experiments, analyzed the data, prepared figures and/or tables, and approved the final draft.
- Weiqian Xu, Chao Teng and Shuangshuang Wang performed the experiments, prepared figures and/or tables, and approved the final draft.

## Human Ethics

The following information was supplied relating to ethical approvals (i.e., approving body and any reference numbers):

All procedures for this study were reviewed and approved by the Institutional Review Boards of the Wenzhou Kangning Hospital (20180412001), and written informed consent was obtained from each participant prior to the performance of any procedures related to this study.

## Data Availability

The raw measurements are available in the Supplementary File.

## Supplemental Information

Supplemental information for this article can be found online at http://dx.doi.org/10.7717/peerj.11332#supplemental-information.

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
