# Peer review of "Association between gonadal hormones and osteoporosis in schizophrenia patients undergoing risperidone monotherapy: a cross-sectional study"

_PeerJ, doi:10.7717/peerj.11332_

## Round 0.1 · original submission · Major Revisions

The study has interesting and clinically relevant findings. However, the paper has major issues to be addressed. Please revise it accordingly. My comments for it are provided as below.

1. Title is problematic because this is not a prospective or interventional study on impact of risperidone on sex hormone and BMD. The current study cross-sectionally assessed the level of BMD and analyzed associated factors of osteoporosis in inpatients with schizophrenia receiving risperidone mono-therapy.

2. English language of the paper is very poor. Please have the paper edit by English-speaking professionals for this revision. For example, line 23, have risk for, line 56, related with, ling 57, resulting in, line 59, ample studies, line 63, in the worldwide, line 75, ample evidences, line 97, SGAs risperidone, line 98, had other diagnosed psychiatric disorder besides.

3. Terms such as “schizophrenia patients” are problematic. Please use person first language to avoid any discrimination against persons with serious mental illness. “Patients with schizophrenia” or “persons living with schizophrenia” is appropriate.

4. Line 30-31, please briefly describe how the relationship between sex hormone and BMD was analyzed.

5. First paragraph of introduction. This is a clinical study, so there is no need to talk so much about the public health significance of osteoporosis.

6. Line 69-70 said the unclear mechanisms of decreased BMD and osteoporosis, but line 70-72 described a lot risk factor of osteoporosis in schizophrenia. These sentences are confusing. Please re-write these sentences.

7. Line 77-81 , please re-organize the two sentences. I can not see the relationship between studies with different antipsychotics and the role of sex hormone in osteoporosis. For introduction, because the study used patients receiving risperidone mono-therapy, the authors may consider to explain the strength of the use of such patients.

8. Line 86, “small sample size” does not belong to “uncontrolled confounding factors”.

9. Line 90-91, the objective of this study is over-stated because the current study design is not an experimental study, which can not answer the mechanisms of osteoporosis regulated by risperidone. In fact, as I mentioned before, this is a cross-sectional study only.

10. Line 103, it is unclear what did the authors mean by “and to minimize the difference in physical exercise”.

11. Participants. Please specify the criteria for selecting healthy controls, how age and sex were matched to patients? This is very important.

12. Line 118-134, the study has two outcomes, BMD and osteoporosis. Please specify which one is the primary outcome.

13. Statistical analysis. t test is not suitable for all comparisons of continuous variables. Those not following normal distribution should be compared with non-parameter methods. Also, some categorical variables need to be compared by using Fisher exact test. Please check these issues. The authors mentioned ANOVA and ANCOVA, but I did not find any results of comparison performed by ANOVA and ANCOVA. For multiple logistic regression, the authors included significant factors associated with BMD. BMD is not binary variable, not suitable for logistic regression. In fact, the statistical analysis was conducted in a confused manner. I think it can be more concise and clear.

14. Tables. The authors mentioned “drinking” and “smoking”. Please specify their collection because subjects are inpatients. It is unlikely that inpatients are allowed to drink and smoke, in general.

Reviewer 1 ·

Basic reporting

This study explored the relationship between serum levels of gonadal hormone and bone mineral density or osteoporosis in patients with schizophrenia treated with risperidone, which is one of the significant clinical issues. This study is well designed with relatively large sample. However, some methodological deficiencies remained. Comments to this manuscript have been listed as follows for quality improvement.

Experimental design

The design is well addressed.

Validity of the findings

The findings are interesting.

Additional comments

Abstract:
-Result: (Bonferroni corrected P's < 0.05) should be modified.

Main text:
There are some small spelling errors and grammars in this article, please pay attention to them. Some of them have been presented as follows.

Introduction:
-Paragraph 5: Please introduce the study hypothesis in this paragraph.

Materials and Methods:
Participants:
- DSM-IV should be added with reference.

-Please polish the following sentence, “They were excluded if they: (1) had other diagnosed psychiatric disorder besides schizophrenia or a lifetime substance abuse/dependence disorder; (2) severe cardiovascular, hepatic, or renal diseases that affect bone metabolism, such as diabetes and hyperthyroidism; (3) were pregnant or breastfeeding; (4) history of bone fracture within one year prior to the enrollment”.

Statistical analysis:
-The author should specify dependent variables and independent variables in stepwise multiple logistic regression analysis .

Discussion:
-Paragraph 3
“The above results in accordance with the phenomenon of high incidence rate of osteoporosis in postmenopausal women and support the role of estrogen involved in bone metabolism”
-This sentence should be polished and some references should be added.

·

Basic reporting

This manuscript is well written.

Experimental design

The experimental design is reasonable and clear.

Validity of the findings

The findings are novel.

Additional comments

This is a well organized study that examined the relationship between serum levels of gonadal hormone and bone mineral density or osteoporosis in schizophrenia patients treated with risperidone. The sample size is large enough. Finally, the authors found a high prevalence of osteoporosis and significant reduced BMD in schizophrenia patients with risperidone monotherapy. This study has great clinical implications. Here are some concerns that may help improve the quality of this manuscript.

1) The main text should start from a separate page from the Abstract.

2) Introduction: the sentence "In the clinical practice, risperidone is a widely used second generation antipsychotic (SGA), and also a prolactin-elevating SGA compound." The cited study was published in 2012. Please update the reference.

3) Introduction: please introduce the study hypothesis, if any.

4) Method: "(1) had been diagnosed with schizophrenia according to the Diagnostic and Statistical Manual of Mental Disorders, Fourth Edition (DSM-IV)" Please double check the diagnostic criteria. In China, the ICD-10 was used in clinical practice.

5) Statistics: please specify dependent variables and independent variables in multiple logistic regression analysis.

6) Discussion: "In addition, we only recruited inpatients with the same diet structure and similar physical intensity, thus, to minimize the disturbance of those factors on our conclusions." This sentence addressed a limitation, rather than strength.

7) There are some minor spelling errors in the text. For example, "SGAare" and "detailedinformation". Please insert space in betwee.

---

## Round 0.2 · Minor Revisions

I am not satisfied with the English language of the paper, despite the substantially improved quality of the manuscript. Please have the paper polished by a professional editor.

Reviewer 1 ·

Basic reporting

none

Experimental design

none

Validity of the findings

none

Additional comments

The authors have addressed my concerns.

---

## Round 0.3 · accepted · Accept

I am pleased to accept this revised version.